# Ka Band Low Channel Mutual Coupling Integrated Packaged Phased Array Receiver Front-End for Passive Millimeter-Wave Imaging

**DOI:** 10.3390/mi14040859

**Published:** 2023-04-15

**Authors:** Xi Chen, Anyong Hu, Jianhao Gong, Jungang Miao

**Affiliations:** School of Electronic and Information Engineering, Beihang University, Beijing 100191, China; chenxi0913@buaa.edu.cn (X.C.); gongjh@buaa.edu.cn (J.G.); jmiaobremen@buaa.edu.cn (J.M.)

**Keywords:** coupling, receiver, multi-channel, integrated package, phased array, passive millimeter-wave imaging

## Abstract

This paper presents a Ka band eight-channel integrated packaged phased array receiver front-end for a passive millimeter-wave imaging system. Since multiple receiving channels are integrated in a given package, the mutual coupling issue affecting the channel will deteriorate imaging quality. Therefore, in this study, the influence of channel mutual coupling on the system array pattern and amplitude phase error is analyzed, and the design requirements are proposed according to the results. During the design implementation, the coupling paths are discussed, and passive circuits in the path are modeled and designed to reduce the level of channel mutual coupling and spatial radiation. Finally, an accurate coupling measurement method for a multi-channel integrated phased array receiver is proposed. The receiver front-end achieves a 28~31 dB single channel gain, a 3.6 dB noise figure, less than −47 dB of channel mutual coupling. Furthermore, the array pattern of the two-dimensional 1024 channel system composed of the front end of the receiver is consistent with the simulation, and the receiver’s performance is verified by a human-body-imaging experiment. The proposed coupling analysis, design, and measurement methods are also applicable to other multi-channel integrated packaged devices.

## 1. Introduction

A passive millimeter-wave (PMMW) imaging system [1,2] uses a high-sensitivity millimeter-wave receiver to obtain the electromagnetic radiation of a target and generate a brightness temperature image of the specified area according to the difference in the radiation characteristics of the measured target. As a millimeter wave can penetrate clothing to image dangerous explosives or contraband items such as metal, powder, and liquid, PMMW imaging technology is an effective technical method for rapid inspection in security applications [3]. Passive image formation relies primarily on radiometric techniques for the reception and focusing of the radiation field generated by the human body. The degree of precision with which a PMMW system can measure radiometric temperature is called radiometric sensitivity. It is the minimum detectable temperature difference that the system can resolve, which is the key index of a passive system. In the past two decades, various imaging systems have been successfully applied, but there are certain limitations, e.g., the fixed nature of the focal plane system’s focal length that limit their imaging distance, the sensitivity and scanning speed of phased array systems are mutually restricted [4]. Moreover, in order to improve imaging sensitivity, spatial resolution, and imaging rates, these systems are being developed toward achieving high integration and large-scale arrays. Beihang University proposed a PMMW imaging system (BHU-1024) for fast and non-cooperative security application scenarios, in which a phased array and a synthetic aperture work synergistically [5,6]. The phased array forms a fan beam and obtains the resolution in the horizontal direction via electric scanning. At the same time, high resolution in the vertical direction is achieved by applying aperture synthesis to each beam so as to obtain the millimeter-wave image within the field of view (FOV). The inspected person walks freely in the security lane, and the system automatically focuses and images at the given video rate. Therefore, the receiver front-end with high sensitivity and high integration is the core of a large-scale array-based imaging system. The system’s architecture is shown in Figure 1.

With the development of Monolithic Microwave Integrated Circuit (MMIC) technology, multi-channel receiver front-end based on COMS or SiGe technology have been widely used in the field of radar and communication [7,8,9,10,11]. However, for PMMW imaging applications, there are still challenges in terms of the full integration of silicon-based phased arrays. Firstly, the high noise figure characteristicdoes not meet the requirements of high-sensitivity systems. Secondly, a problem concerning channel mutual coupling caused by high integration will be encountered. Therefore, we chose to use a multi-function receiver chip based on the GaAs process and integrated packaging technology to develop an eight-channel phased array receiver front end [12,13,14,15,16]. At the present stage, this design scheme is a reasonable choice for realizing such a large-scale array system considering radiometric sensitivity performance, development cost, and design risk.

In this paper, an integrated packaged receiver front-end framework for a PMMW imaging system is proposed. The channel mutual coupling issue caused by high integration is emphatically analyzed and discussed with respect to three aspects of array architecture, circuit design, and measurement methods. The main contributions of this paper can be summarized as follows.

(1) Based on the system’s array structure, the Radio Frequency (RF) and Local Oscillator (LO) coupling paths are modeled and analyzed. According to the relationship between the deterioration of the array pattern and phase-shifting errors caused by channel mutual coupling and the radiometric sensitivity of the PMMW imaging system, the requirements for RF coupling and LO coupling are proposed to be less than −25 dB and −20 dB, respectively. The above results provide guidance for the subsequent design.

(2) With the aim of reducing channel mutual coupling and spatial radiation, transmission line, bonding interconnection, and LO power divider structures are designed through modeling and simulation.

(3) By exploiting the characteristic that each channel of a phased array can adjust its phase independently, a channel-mutual-coupling measurement method is proposed, which affords higher measurement accuracy compared with the previous two-channel measurement method. The degree of channel mutual coupling measured by this method is less than −47 dB, and this method is also suitable for other multi-channel devices.

## 2. Analysis of Channel Mutual Coupling on a Linear Phased Array

### 2.1. Receiver Front End Array Architecture in the System

As can be seen from the receiver front end array architecture of the PMMW imaging system presented in Figure 1, the system consists of 32 linear phased arrays, and each phased array contains 32 receiving channels. According to the target information fed back by the distance sensor, the system tracks the movement of the target at 1~5 m in front of the antenna array aperture and performs real-time imaging at the rate of a video. The imaging range is set to be 1 m wide and 2 m high, which covers the size of one person. Therefore, the largest imaging FOV is in the horizontal direction at 1 m in front of the array, which corresponds to $\varphi_{p}=\pm arctan\left(0.5/1\right)=\pm 26.5{}^{\circ}$. For a linear phased array, as shown in Figure 2, when the grating lobe enters the FOV, the radiation energy in the grating lobe’s direction is also received by the array, which reduces the main beam efficiency, thus deteriorating the radiometric sensitivity of the system. Therefore, the FOV also needs to be defined as the range without grating lobes [17]. The grating lobe’s position can be determined from the maximum value of the linear array pattern:(1)$$2\pi \frac{d_{x}}{\lambda }\left(sin\varphi_{p}-sin\varphi_{0}\right)=2\pi$$
where the wavelength λ is 8.8 mm (the operating frequency of the system is 32~36 GHz). When the beam is focused on the edge of the FOV, the grating lobe appears on the other side. Thus, $\varphi_{p}$ and $\varphi_{0}$ are both equal to ±26.5°, and considering the processing difficulty and array layout, the receiver front end is designed as an eight-channel integrated structure, so the final channel spacing is $d_{x}=10\;mm$. That is to say, each phased array contains four eight-channel receivers, and the maximum aperture of the array is $D_{x}=418\;mm$, including the space between modules for support and heat dissipation. The spatial resolution defined by the half-power beamwidth is expressed as:(2)$$∆\theta_{x}=0.88\frac{\lambda }{D_{x}}$$

The calculated spatial resolution of the array is about 1.06°, which is 1.85 cm at 1 m in front of the array aperture. This size is sufficient to detect the majority of dangerous explosives or contraband items.

### 2.2. Influence Analysis of Channel Mutual Coupling on the Phased Array Pattern

Figure 2 shows a simplified block diagram of a linear phased array. Such an approach was chosen to indirectly change the phase of the 32~36 GHz received signal by adjusting the phase of the LO signal [18]. When channel mutual coupling exists, coupling matrix C should be considered in the calculation of the pattern:(3)S=C·A
(4)$$C=\left[\begin{array}{cccc} C_{m} & 0 & 0 & 0\\ 0 & C_{m} & 0 & 0\\ 0 & 0 & C_{m} & 0\\ 0 & 0 & 0 & C_{m} \end{array}\right]$$
(5)$$C_{m}=\left[\begin{array}{ccccc} 1 & c_{12}e^{j\varphi_{12}} & c_{13}e^{j\varphi_{13}} & \cdots & c_{18}e^{j\varphi_{18}}\\ c_{12}e^{j{-\varphi }_{12}} & 1 & c_{23}e^{j\varphi_{23}} & \cdots & c_{28}e^{j\varphi_{28}}\\ \vdots & \cdots & & \ddots & \vdots \\ c_{18}e^{j{-\varphi }_{18}} & c_{28}e^{j-\varphi_{28}} & c_{38}e^{j-\varphi_{38}} & \cdots & 1 \end{array}\right]$$
where matrix $A\in {∁}^{32\times 1}$ is the vector of each single channel, which is described as the wave path-difference from each channel to the beam-focusing position and the phase change during electrical scanning; meanwhile, the amplitude of each channel is assumed to be consistent. The mutual coupling of a 32-channel phased array is represented by $C\in {∁}^{32\times 32}$. However, each set of eight channels is integrated into a package, and coupling only occurs within the package, so the degree of coupling between different packages is 0, which simplifies the coupling matrix C. $C_{m}$ represents the mutual coupling between different channels in the same package. The mutual coupling of the adjacent channel can be considered as having the same amplitude and an opposite phase. Therefore, in Equation (5), $c_{nm}$ and $\varphi_{nm}$ represent the coupling coefficients of channel n and channel m, in which $c_{12}$ and $c_{21}$ are the same, $\varphi_{12}$ and $\varphi_{21}$ are opposite, and $c_{11}$~$c_{88}$ are all 1, thereby resulting in a $C_{m}\in {∁}^{8\times 8}$ matrix.

According to the previous system’s structure, the size of the 32-channel phased array is $D_{x}=418$ mm, and the imaging distance is 1–5 m, which is much smaller than $2D^{2}/\lambda =39.71\;m$. Therefore, for the PMMW system, the imaging range is in the near-field of the phased array, and the near-field pattern is simulated and calculated at 2 m in front of the antenna array, as shown in Figure 3. In the figure, (a) Figure 3a is the normalized pattern of beam focusing at different positions within the FOV under a 32~36 GHz broadband signal. It can be seen that the gain of the pattern focusing on the edge of the FOV decreases and the half-power beamwidth widens. Figure 3b shows some examples of instances when coupling is considered, for which the beam-focusing position is at x = 0.3 m. The channel coupling amplitude $c_{nm}$ is randomly selected within the range of −5 ± 3 dB, −15 ± 3 dB, −25 ± 3 dB, and −35 ± 3 dB, and the phase $\varphi_{nm}$ is selected in the range of 0~360°. Due to the randomness of the coupling value, the results of the presented pattern with coupling are not unique. The existence of coupling will cause high side lobe in the FOV and interfere with imaging results.

When the value of the coupling coefficient is different, the obtained pattern is also different. Therefore, in order to quantify the influence of coupling on the pattern, the changes in main lobe width, beam direction, side lobe level, and beam efficiency are calculated and counted by repeating random values several times within the set coupling ranges. Figure 4 shows the average value of the simulation results repeated 500 times. Figure 4a is the statistical result of the half-power beamwidth of the pattern. The curve results were consistent at different coupling levels, indicating that coupling had little effect on the main lobe. It also can be seen that the beamwidth increases when the focus position is at the edge of the FOV, which is consistent with the results of the patterns in Figure 3a. Figure 4b shows that when no coupling occurs, a beam-focusing position error also exists. This is because when a phase shift occurs in the LO link, only the center frequency of 34 GHz is used as a reference to calculate the wave path difference between different channels to the beam-focusing position. When different frequencies are shifted by the same phase, a focusing position error will occur. Although it is compensated in the broadband range, when pointing to the edge of the FOV, there will still be some deviation due to the asymmetric paths of different channels in the array. However, the influence of channel mutual coupling on the beam-focusing position error is not obvious. It can be seen from Figure 4c,d that the sidelobe level and main beam efficiency are affected by coupling, and the decrease in beam efficiency will deteriorate the radiation sensitivity of the imaging system. In general, when the level of coupling is less than −25 dB, the above indicators are all approximately equivalent to those without coupling, thereby stipulating design requirements for the phased array receiver front end.

It can be clearly seen from the simulation above that as the level of coupling increases, the array pattern’s beam efficiency deteriorates. For the PMMW imaging system, the core indicator of radiometric sensitivity is directly proportional to the beam efficiency. For quantitative analysis, it is necessary to exhaustively calculate these values. A curve depicting the change in beam efficiency with respect to the coupling level is shown in Figure 5. When the degree of coupling is greater than −25 dB, beam efficiency begins to decrease, which means that the minimum distinguishable brightness temperature of the system decreases. Therefore, for an imaging system with a fixed array structure, it is reasonable to set the mutual coupling of RF channels to less than −25 dB in order to achieve optimal sensitivity.

### 2.3. Influence Analysis of Channel Mutual Coupling on LO Amplitude and Phase Error

Figure 6 shows the coupling path in the phased array LO link, and the existence of this coupling will affect the amplitude and phase changes of each channel’s LO signal.

The first path is the degree of coupling between the channels following the output of the LO power divider, which can also be expressed by Equation (3). The same method was used to conduct random, repeated experiments under different coupling magnitudes. The distribution of amplitude and the phase errors caused by coupling are shown in Figure 7a,b. Figure 7c,d present the results calculated using the cumulative distribution function (CDF). When the degree of coupling is −25 dB, there is a 90% probability that the amplitude error does not exceed 1.5 dB and the phase error is less than 10°. The simulation results provided us with the following insights. Firstly, when designing the receiver front end, set the driving power such that it is high enough to allow the frequency multiplier or mixer on the LO link to function in the saturation state so that the 1.5 dB change in the LO signal will not affect the amplitude of the RF path. Secondly, since the circuit and mechanical structure are fixed, the phase error caused by the LO channel’s mutual coupling will only affect the initial phase value of each channel. When the phased array is focused, the error can be calibrated by the system. Therefore, the phase error caused by this coupling path does not affect the operation of the phased array.

The second kind of LO coupling path occurs when the reflected signal, after passing through the LO phase shifter, is coupled with other channels through a power divider. The phase change caused by this coupling is related to the phase shift of other channels, so it exists in real time during beam scanning and cannot be calibrated. Taking the two channels in the figure as an example, the above process can be described as the following expression:(6)$$S_{1}=A_{1}+\Gamma_{1}\cdot A_{2}{\cdot C}_{21}+\Gamma_{2}\cdot A_{2}{\cdot e^{-j2*\phi_{PS}}\cdot C}_{21}$$
(7)$$A_{1}=a_{1}e^{-j\phi_{1}},{A_{2}=a}_{2}e^{-j\phi_{2}}$$
(8)$$C_{21}=c_{PD21}e^{-j\phi_{PD21}}$$
where Γ is the reflection coefficient of the device port, which can be assumed to be equal to −15 dB in reference to the chip design parameter. $C_{21}$ is the degree of coupling between the output ports of the power divider, also known as isolation. Similarly, arbitrary values are selected in different coupling amplitude ranges for analysis. Figure 8a shows the channel 1 phase error caused by coupling when channel 2 phase shifting occurs. Figure 8b presents the statistics of the maximum amplitude phase error caused by different coupling magnitudes. It can be seen that when the level of port coupling of the power divider is less than −20 dB, the resulting LO amplitude error is 0.2 dB, which has no impact on the RF signal. The resulting LO phase error is 0.7°, and since the phase shift occurs before the quadrupler is activated, the phase shift error at the RF is 2.8°, which will affect beam focusing and the scanning of the phased arrays. It also reduces beam efficiency, which, in turn, worsens radiometric sensitivity. Statistics were calculated according to different phase shift errors, and the results are shown in Figure 9a. A phase shift error of 2.8° has little effect on beam efficiency, so −20 dB of LO power divider port coupling is acceptable.

Further, in order to facilitate the comparison of the impact of the RF and LO coupling amplitudes on system sensitivity, we plotted the curves of the variance of beam efficiency with the RF and LO coupling levels in Figure 9b. When the RF and LO coupling levels reach the order of −10 dB, the corresponding beam efficiency level decreases by 2% and 8%, respectively. Therefore, RF channel coupling requires further attention.

## 3. Phased Array Receiver Front End and Suppression Design of Channel Mutual Coupling

### 3.1. Phased Array Receiver Front End

The input ports of the phased array receiver front end consist of eight independent horn antennas, which are used to receive 32~36 GHz millimeter-wave signals. The 10 GHz LO input and the 4~8 GHz Intermediate Frequency (IF) output signals are distributed or synthesized by the power dividers in the circuit. The active circuit in each channel is only composed of two highly integrated, self-developed, multi-functional receivers, namely, an MMIC and a hybrid MMIC, which have low noise amplification, down conversion, sideband separation, frequency doubling, analog phase shifting, and gain adjustment functions. The frequency conversion gain of these two cascaded MMICs is 23 dB, and their gain adjustment range is 8 dB. When the control voltage varies from 0 to 1.5 V, the phase shift range is greater than 360°. These characteristics show that these MMICs are suitable for the beamforming of the phased array. The noise figure is about 3 dB, and the image rejection ratio is greater than 35 dB, which indicates that the MMICs have low noise characteristics and meet the requirements for a high-sensitivity receiver. The block diagram structure of the eight-channel phased array receiver front end is shown in Figure 10.

A double-layer planar circuit layout method was adopted to implement an integrated packaging design. The front side is a millimeter-wave circuit including multi-functional MMICs and a multilayer Printed Circuit Board (PCB) integrating IF, a power supply, and control circuits. The back side is the LO circuit, which transmits the LO signal to the front receiver MMIC through a coaxial transition structure in each channel. Finally, the horn antenna and waveguide transition are integrated with the shielding box. Photos of the receiver front end are shown in Figure 11.

### 3.2. Influence Analysis and Suppression Design of Channel Mutual Coupling in RF and IF Paths

According to the results of the influence of channel mutual coupling on the array presented in Section 2, it was necessary to analyze the possible coupling paths and suppress them when designing the receiver. In Figure 12, taking the two channels as an example, the signal $V_{1}$ is input from channel 1, and coupling may initially occur on the RF path α; secondly, due to the high integration and compact size of the MMICs, $V_{1}$ may be coupled to channel 2 through path β; and thirdly, the signal may also be coupled on the IF path γ before entering the power combiner. The outputs of the two channels can be expressed as follows:(9)$${V_{1}}^{\prime }={GV}_{1}e^{-j\theta_{1}}$$
(10)$${V_{2}}^{\prime }=C_{\alpha }e^{-j\phi_{\alpha }}{GV}_{1}e^{-j\theta_{2}}+C_{\beta }e^{-j\phi_{\beta }}V_{1}+C_{\gamma }e^{-j\phi_{\gamma }}{GV}_{1}e^{-j\theta_{1}}$$
where G is the gain of the cascade MMICs (equal to 23 dB), θ is the phase shift of the channel, and $C_{\alpha }$, $C_{\beta }$, and $C_{\gamma }$ represent the coupling coefficients of the three paths. The second term in Equation (10) is the coupling of the RF signal with IF. Due to the large frequency difference and the lack of gain, G, the influence on ${V_{2}}^{\prime }$ prime is small compared with the other two terms and can thus be ignored. The remaining $C_{\alpha }$ and $C_{\gamma }$ can be considered as the degrees of coupling on the RF and IF paths, respectively.

The RF input terminal is a combined structure, consisting of an end-fed fin-line and a ridge waveguide, that transmits the millimeter-wave signal from the horn antenna to the planar microstrip circuit, thus meeting the array structural requirement and the broadband performance requirement. Two-channel microstrip and coplanar waveguide-with-ground (CPWG) transmission line models were established, as shown in Figure 13a,b. The substrate is Rogers 3003 with a thickness of 10 mil, a length of 16 mm, and a channel spacing of 10 mm. When there is metal shielding wall between the channels, it is considered that no coupling is occurring. However, during assembly, there may be gaps between the metal wall and the cover plate, which cannot be completely shielded. Assuming that the gap is 0.1 mm, the coupling level $C_{\alpha }$ is about −50 dB, which is slightly lower than that without the metal wall. In the same state, when changing the microstrip line to CPWG, the simulation result of the coupling amplitude can drop below −60 dB, and these simulation results are shown in Figure 13c,d. Therefore, when the circuit is designed, metal shielding walls are added between the RF transmission paths and channels, while CPWG transmission lines are selected. The metal wall and the shielding box are processed as one, which will not increase the processing cost.

The CPWG transmission line was also used for an IF circuit. In the absence of a metal wall, it can be seen from the comparison of different lengths through simulation that the coupling level $C_{\gamma }$ is lower than −80 dB when the transmission line length is 5 mm. The simulation model and results are shown in Figure 14. These results show that IF output of the eight receiving channels should be closely connected with the IF combiner, making the interconnecting transmission lines as short as possible. At this time, the degree of channel mutual coupling in the RF and IF transmission paths mainly depends on $C_{\alpha }$.

According to the above results, it can be understood that the CPWG transmission line makes it easier for the electric field to be bound between the transmission conductor and the ground metal, so it radiates into space to a lower degree than the microstrip line and, consequently, the degree of mutual coupling between channels is smaller. However, in addition to the uniform transmission line in the RF path, there are also bonding wire structures connected with the MMICs. Through an electromagnetic simulation, it was found that the electric field near the bonding wire is more likely to radiate into space, causing coupling between channels. Therefore, two kinds of bonding wire structures were designed and compared [19]. The first one is a matching compensation structure, in which the transmission line on the PCB is only connected to the signal pad of the MMIC, and the inductive parasitic effect introduced by the bonding wires is compensated by designing a multi-level-matching scheme for use on the PCB. The second structure entails connecting the signal and ground of the CPWG transmission line to the Ground–Signal–Ground (GSG) pad on the MMIC to afford a return path that minimizes impedance mismatch. The models and the full wave simulation are shown in Figure 15.

In order to analyze the radiation characteristics of the discontinuous interconnection structure, we selected the edge position of the transmission line where the bonding wires are connected to the PCB as a reference and simulated the decreasing trend of the electric field intensity with the increase in height from the circuit surface. The simulation results are listed in Figure 16a. It can be seen that the electric field intensity of the CPWG near the circuit’s surface is stronger than that of the microstrip line, but it decreases faster with the height rising. When the height is greater than 0.5 mm, the electric field intensity is smaller than that of the microstrip line, indicating that the degree of radiation to space is small, which confirms the conclusion regarding a small degree of coupling of CPWG that was presented in Figure 13d. Similarly, when comparing the simulation results of the two kinds of bonding wires, the electric field intensity radiated by the GSG structure is less than that of the compensation structure. Although the return loss in the 32~36 GHz range is not as good as that of the compensation structure, it is also less than −16 dB, as shown in Figure 16b. Therefore, from the perspective of preventing coupling, the assembly method of bonding wire in GSG form is selected.

### 3.3. Influence Analysis and Suppression Design of Channel Mutual Coupling in LO Path

Since the LO frequency of 10 GHz is close to the IF and the channel spacing is also 10 mm, the coupling between LO channels can be ascertained by referring to the results presented in Figure 14b, which present a level lower than −50 dB. In addition, according to the analysis provided in Section 2, the amplitude and phase errors caused by the coupling on the LO transmission path can be calibrated. Therefore, more attention was paid to the coupling occurring on the LO power divider.

The LO eight-way Wilkinson power divider has a two-stage structure [20]. Compared with the output return loss, isolation is optimized preferentially, and the values of the isolation resistors are 82 Ω and 300 Ω. Further, isolation grounding is added around the transmission path, the layout of which is shown in Figure 17a. The degree of isolation is obtained by measuring the insertion loss of any two output ports. As P1–P2 are adjacent output ports, there is only one stage isolation resistor between them, so their isolation is minimum; for the same reason, P1–P3 or P1–P5 will have higher degrees of isolation. In Figure 17b, the minimum isolation is greater than 22 dB, thus meeting the LO coupling suppression requirements.

## 4. Measurement of the Receiver Front End and Array

### 4.1. Channel Mutual Coupling Measurement Method of Multi-Channel Integrated Package

Due to the high integration level, each channel cannot be separated, and there is only one output port, so the degree of mutual coupling between channels cannot be directly measured by using traditional methods. Therefore, a new channel mutual coupling measurement and calculation method is proposed by exploiting the ability of the phase of each channel in the receiver to be adjusted independently. According to the analysis presented in Section 3.2, $C_{\alpha }$ mainly determines the coupling amplitude in the RF and IF paths. Taking a four-channel model as an example, as shown in Figure 18a, the input signal Vin of channel 1 will be coupled to other channels on the RF path. Therefore, in addition to the output signal $V_{1}$, other channels also have outputs $V_{2}$, $V_{3}$, and $V_{4}$, which are synthesized inside the circuits, and only Vout can be measured. Coupling occurs before the input of the phase shifter, so the phase can be changed by controlling the phase shifter of the coupling channel, and since the package structure is fixed, it can be assumed that the coupling coefficient of each channel remains unchanged during the phase-shifting process. In this way, when using the rotating element electric field vector method to change the phase of channel 2 [21], the change in the amplitude of the output signal can be observed, as shown in Figure 18b. Specifically, the vector sum of the four-channel output signals is the Vout that can be measured. When only the phase of channel 2 is changed from around 0 to 360°, the change path of the Vout is a circle. The radius of the measured circle is the modulus of the coupling signal $V_{2}$. The center of the circle is the modulus of the vector sum of the other channels, which is approximately equal to the modulus of $V_{1}$ when the level of coupling is small. When the phase of $V_{2}$ is consistent with the vector sum of the other three channels, the maximum output value ${Vout}_{max}$ is obtained, while the minimum value ${Vout}_{min}$ is obtained when $V_{2}$ is inverted. Based on the maximum and minimum values in the measurement results, the output amplitudes of the measured channel and coupling channel can be calculated. The mutual coupling of channels 1 and 2 is the result of their division. The expression corresponding to this mutual coupling measurement method is as follows.
(11)$$\left|V_{1}\right|\approx \left|V_{1}+V_{3}+V_{4}\right|=\frac{\left|{Vout}_{max}+{Vout}_{min}\right|}{2}$$
(12)$$\left|V_{2}\right|=\frac{\left|{Vout}_{max}-{Vout}_{min}\right|}{2}$$
(13)$$c_{12}=\frac{\left|V_{2}\right|}{\left|V_{1}\right|}$$

This method is more suitable for two-channel models [22] because when calculating the $\left|V_{1}\right|$ of a multi-channel model, an approximate value is used, for which there will be calculation error. In order to obtain a more accurate $\left|V_{1}\right|$, it is necessary to add a measurement step, that is, to shift the phase of channel 1. In this process, the vector sum of the coupling signals is unchanged, and since the output signal $\left|V_{1}\right|$ is larger, the vector superposition diagram corresponds to the relationship shown in Figure 18c. The $\left|V_{1}\right|$ value obtained is more accurate and can be expressed as:(14)$$\left|{V_{1}}^{\prime }\right|=\frac{\left|{{Vout}^{\prime }}_{max}+{{Vout}^{\prime }}_{min}\right|}{2}\;,\;{c_{12}}^{\prime }=\frac{\left|V_{2}\right|}{\left|{V_{1}}^{\prime }\right|}$$

The above two calculation methods were compared by simulating an eight-channel integrated package. The coupling amplitude of each channel varies from −50 dB to −15 dB, and the phase corresponds to a random value from around 0 to 360°. Since the influence of phase on the result of the vector sum is accidental, 500 simulations were carried out, and the average and maximum values were statistically analyzed. The results are shown in Figure 19. It can be seen that the results calculated using the optimization method have no deviation from the set coupling coefficient, while the results calculated using the approximate method increase with the magnitude of coupling. When the level of coupling is lower than −30 dB, an approximate method can also be used, and thus the maximum error is less than 1.5 dB.

Since the RF input terminal of the receiver front end integrates the horn antenna with the RF channel, to evaluate single-channel performance, a test fixture was designed with a shape that matches the cavity of the horn antenna, and its interior corresponds to the same waveguide as the measured channel. During measurement, the test fixture was inserted into the antenna and connected to the instrument through the waveguide coaxial transition. Diagrams of the connection mode and test fixture are shown in Figure 20.

Figure 21a shows the channel mutual coupling measurement platform, which uses the receiver mode of the Vector Network Analyzer (VNA N5227B). When measuring, the test fixture is inserted into one of the channels, changing the phase of all channels sequentially to record the change in the IF output power. The measured results of the output power change with the control phase are sinusoidal curves, and the measurement results with channel 1 as the input are shown in Figure 21b. The maximum and minimum values were selected from the curves, and the level of channel mutual coupling was calculated according to Equations (12)–(14). All the results are listed in Table 1, and the maximum coupling observed was equal to −47 dB.

Since the input and output frequencies are different, the transmission phase of the frequency conversion circuits cannot be directly measured. Therefore, in order to measure the channel phase error caused by LO coupling, it is necessary to introduce a reference and use the vector mixing mode of the VNA to measure the relative phase. The measuring platform is shown in Figure 22, in which the output voltage of a Digital to Analog chip (DA) is changed through the control board, thus changing the phase of the analog phase shifter of the measured channel. The control board is used to shift the phase of the other channels, except the channel to be measured, and the channel phase in the reference remains unchanged. The result is the phase error caused by LO coupling during the phase shifting of other channels.

First, the measurement error was calculated, and the phase measurement results were recorded 200 times. In Figure 23a, the measurement results vary within ±0.4° and the standard deviation is 0.14°. Figure 23b shows the phase error measurement results, where channel 1 is the measured channel and the other channel phases change sequentially. It can be seen that the maximum phase error caused by other channels in the phase-shifting process is about 2°, which is close to the analytical results presented in Section 2.3.

The above results prove that the mutual coupling of the RF and LO channels in the eight-channel integrated packaged receiver front end is well suppressed.

### 4.2. Basic Performance of the Receiver Front End

The conversion gain measurement results of each channel are shown in Figure 24a. The in-band gain range is 27.9~31.3 dB, and the single-channel gain result presents a 9 dB decrease in the IF of the eight-channel power combiner. The tuning voltage of the variable gain amplifier can be changed by adjusting the potentiometer such that the gain of each channel is consistent. After adjustment, the channel consistency can reach 0.5 dB. The trend in the single-channel gain with respect to frequency is basically consistent with the measurement results of the receiver and IF cascaded MMICs and is only slightly reduced at 32 GHz (IF 8 GHz) by the influence of the combiner and low-pass filter on the IF. In the same state, the image rejection ratio is greater than 35 dB, as shown in Figure 24b. In Figure 24c, when the control voltage changes from 0.3 to 0.9 V, the phase-shifting range is continuously adjustable, and the range is greater than 360°. The phase-shifting accuracy depends on the accuracy of the voltage configured by DA. The noise figure of the receiver front end was measured using the Y factor method. An absorbent material at room temperature was used as a heat source (Th = 300 K), and an absorbent material immersed in liquid nitrogen was used as a source of cooling (Tc = 80 K). The IF output integrated power Ph and Pc under the heat and cooling sources were measured using a signal analyzer (FSVR40), and the Y factor was calculated to be 1.46; thus, the receiver noise figure was 3.62 dB (Figure 24d).

By utilizing the phase shifts’ characteristic curves, the initial phase of each channel can be calibrated, and the far-field pattern can be scanned. The measurement platform is shown in Figure 25a, where the left side is the transmitting antenna and the right side is the tested receiver, located above the mechanical turntable. The transmitting antenna is 2.5 m in front of the positive radiation direction of the receiver front end, and the distance is much greater than $2D^{2}/\lambda$. The results showed that a single eight-channel receiver met the far-field conditions. During measurement, the turntable was rotated to scan the range from −80° to +80°. The resulting normalized pattern was very consistent with the simulation, as shown in Figure 25b. The main lobe width was 5.6°, while the grating lobes were located at 56.6° and −57.8°.

The above measurements verify that the basic functionality of the eight-channel receiver front end meets the application requirements.

### 4.3. System Integration and Imaging Experiment

The assembled PMMW imaging system is shown in Figure 26a. Above is a 1024-channel array composed of the receiver front end, whose structure was described in Section 2.1. Below are two sets of complex correlation-processing subsystems. The spatial resolution of the system was verified by measuring the half-power beamwidth of the array pattern. The measurement environment is shown in Figure 26b.

Before near-field pattern scanning was conducted, the phased array needed to be beam-focus-calibrated using the rotational vector method, and the required phase configuration in the beam-scanning process needed to be calculated according to the coordinate relationship between the transmitting antenna and the array channel. The measurement and simulation results of a normalized pattern at 1 m in front of the array are shown in Figure 27a. Since the measured point source is a broadband, small-signal noise source, the noise signal received during pattern measurement was weaker when the sidelobe was scanned. In addition, the dynamic range of the direct output voltage of the detector in the system was relatively small, and small fluctuations in the measurement results may have also had an impact on the results. So, compared to the simulation results, the sidelobe in the pattern has a larger error. However, the results near the main lobe are comparable, as shown in Figure 27b. The main lobe width is 1.9 cm, which is consistent with the simulation and meets the system design requirements.

Further, the same method was used to measure resolution at different imaging distances in front of the system, and the results are shown in Table 2. It can be seen that as the imaging distance increases, the resolution also gradually increases; however, within the range of 1–5 m, the resolution remains at the centimeter level.

In the security-oriented human-body-imaging experiment conducted in this study, the inspected person passed through the front of the PMMW imaging system at a normal walking speed, and the system obtained a video result with a frame rate of 25 Hz. Screenshots of the system software are shown in Figure 28a, including visible light video, millimeter-wave image video, and an alarm display, where the imaging results were adaptively processed based on deep learning [22,23]. Figure 28b shows the original millimeter-wave brightness temperature image of the metal gun model carried by the inspected person. The displayed horizontal direction is 1 m wide, and the vertical direction is 2 m high. The contour of the human body in the millimeter-wave image is basically consistent with the visible light image, and the metal gun model in the bag located at the abdomen of the body is also clearly visible. The magnitude value of color shading is the measured brightness temperature, and the brightness temperatures displayed by the gun model and human body are 160 K and 260 K, respectively, corresponding to a difference of 100 K, which is easily distinguishable.

The channel mutual coupling suppression of the high-sensitivity receiver front end lays the foundations for acquiring fast and clear millimeter-wave image results while improving system integration.

A comparison of this study with previously published phased array receiver front end is shown in Table 3. It can be seen that the receiver front end based on CMOS and SiGe technology have the characteristics of high integration and low power consumption. Therefore, they are mostly implemented in the form of a single chip and used in the communication field. However, since the receiver in this paper is applied to PMMW imaging, more attention was paid to the noise figure. It can be seen that the noise figure can still reach 3.6 dB even when the transmission loss of the front stage is included. Meanwhile, due to the higher complexity of the integrated package, it has more advantages in gain, isolation, and phase shift accuracy.

## 5. Conclusions

In this paper, an eight-channel integrated packaged phased array receiver front end was proposed for a PMMW imaging system. Each channel allows for low-noise amplification, down conversion, frequency multiplication, phase shifting, and gain adjustment. First, the influence of mutual coupling between channels on the array pattern and LO phase shifting error was analyzed according to the system’s array layout; then, in the design of the receiver front end, the possible coupling paths of the transmission line, bonding wire interconnection, and power divider were analyzed and channel mutual coupling was suppressed; and finally, for the coupling measurement of the multi-channel integrated packaging, a more accurate method was proposed by using the phase-adjustable characteristics of each channel, and the measured level of channel mutual coupling was less than −47 dB. In general, the eight-channel receiver front end has a single-channel gain of 28~31 dB, a 3.6 dB noise figure, and a continuous phase shift capacity of more than 360°. These results satisfy the high-sensitivity requirements of a PMMW imaging system and lay the foundations for large-scale array integration. The proposed coupling analysis, design, and measurement methods are also worth popularizing for use in other high-integration devices.

## Figures and Tables

**Figure 1 micromachines-14-00859-f001:**
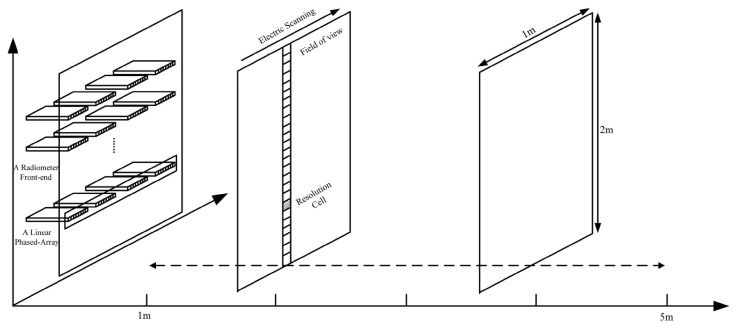
Receiver front end array architecture.

**Figure 2 micromachines-14-00859-f002:**
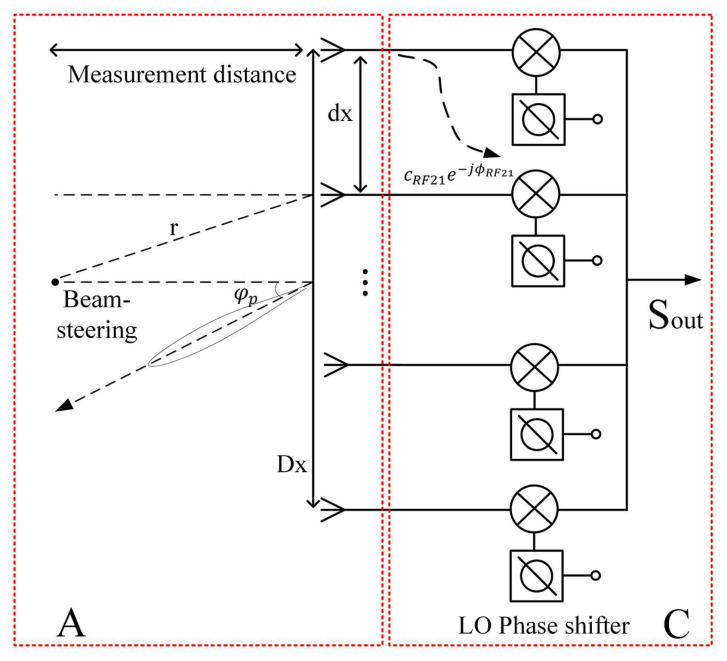
Simplified block diagram of a linear phased array with RF coupling between channels.

**Figure 3 micromachines-14-00859-f003:**
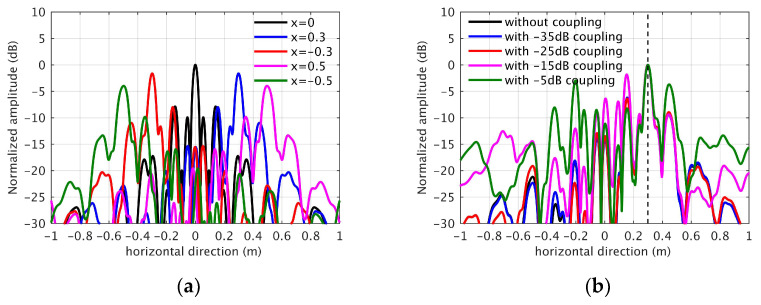
(**a**) Broadband-normalized pattern of beam focusing at different positions; (**b**) examples of patterns at different coupling levels, for which the beam-focusing position is at x = 0.3 m.

**Figure 4 micromachines-14-00859-f004:**
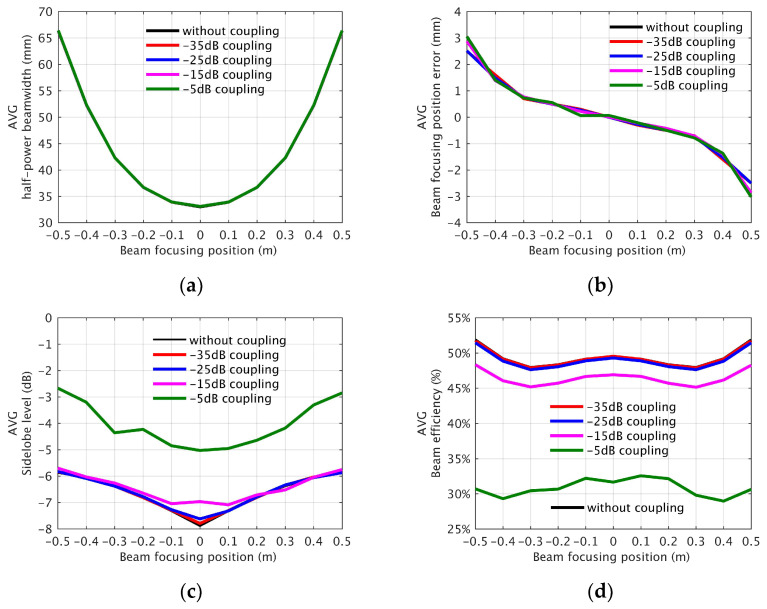
The mean values of repeated simulation results of beam focusing at different positions: (**a**) half-power beamwidth; (**b**) beam-focusing position error; (**c**) sidelobe level; (**d**) main beam efficiency.

**Figure 5 micromachines-14-00859-f005:**
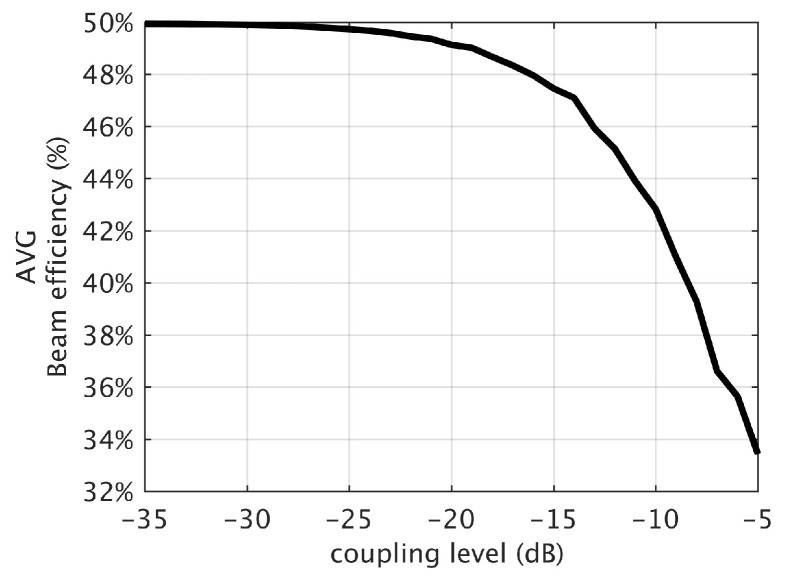
Beam efficiency varies with the coupling level.

**Figure 6 micromachines-14-00859-f006:**
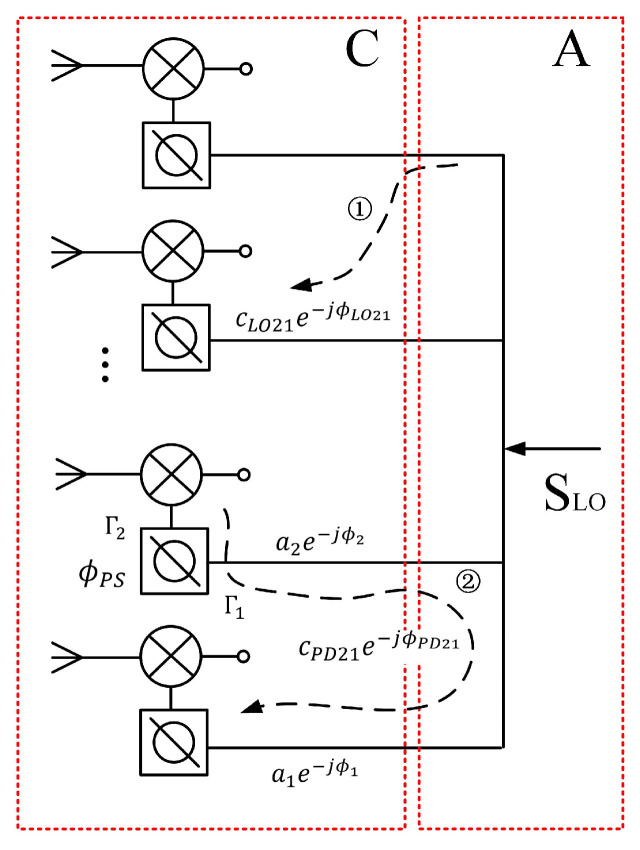
Simplified block diagram of phased array with two different LO coupling paths.

**Figure 7 micromachines-14-00859-f007:**
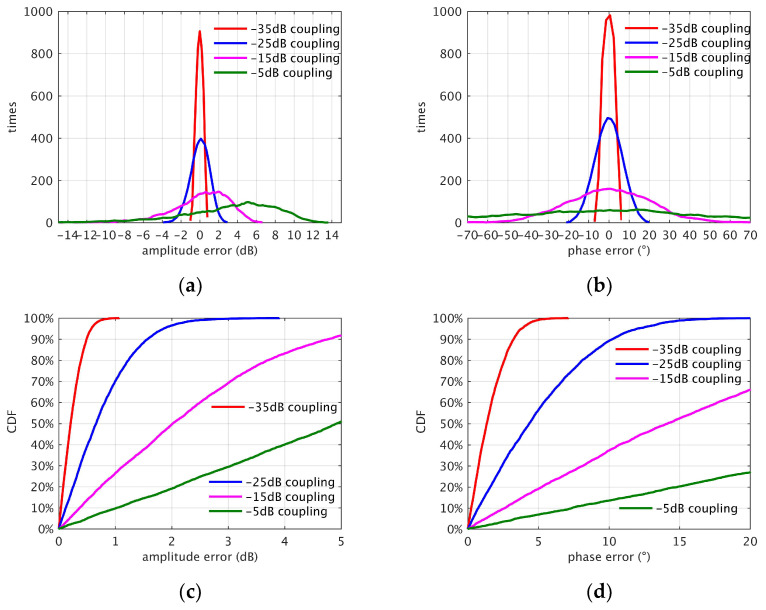
(**a**) Statistical distribution of amplitude error under different coupling magnitudes; (**b**) statistical distribution of phase error; (**c**) cumulative distribution of amplitude error; (**d**) cumulative distribution of phase error.

**Figure 8 micromachines-14-00859-f008:**
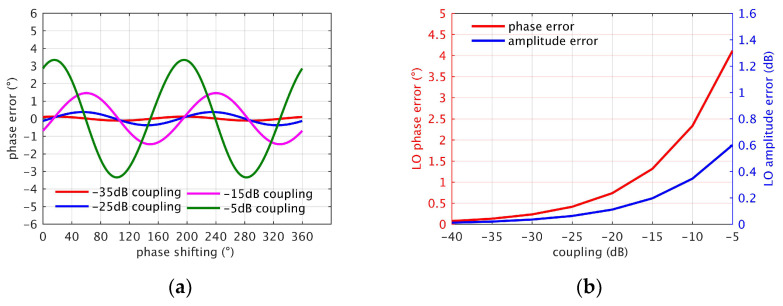
(**a**) Adjacent coupling channel’s influence on phase error related to phase shift; (**b**) statistics of influence on amplitude and phase of adjacent channels during phase shift.

**Figure 9 micromachines-14-00859-f009:**
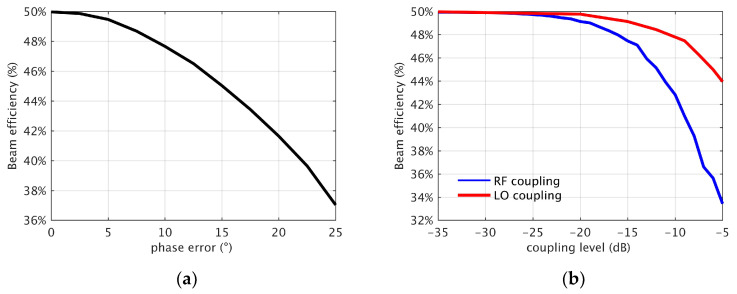
(**a**) Variance of beam efficiency varies with phase error; (**b**) variance of beam efficiency with coupling level, and a comparison of the effects of RF and LO coupling.

**Figure 10 micromachines-14-00859-f010:**
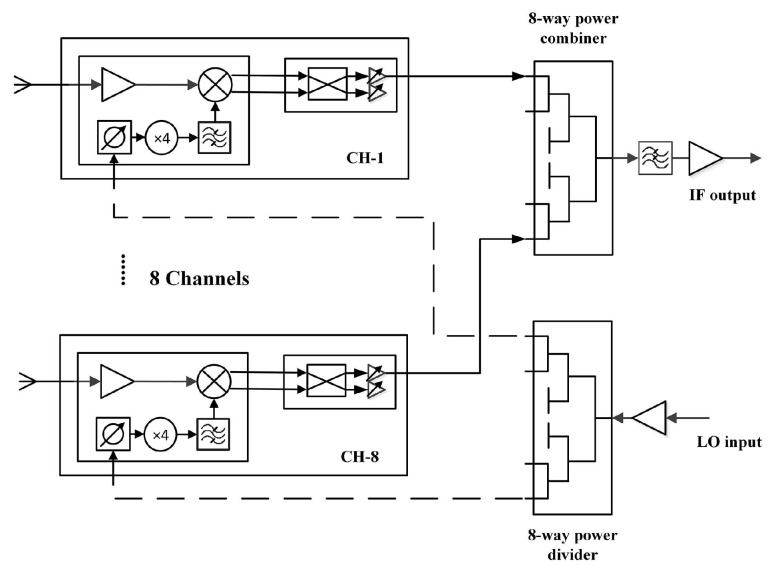
Schematic of the proposed phased array receiver front end.

**Figure 11 micromachines-14-00859-f011:**
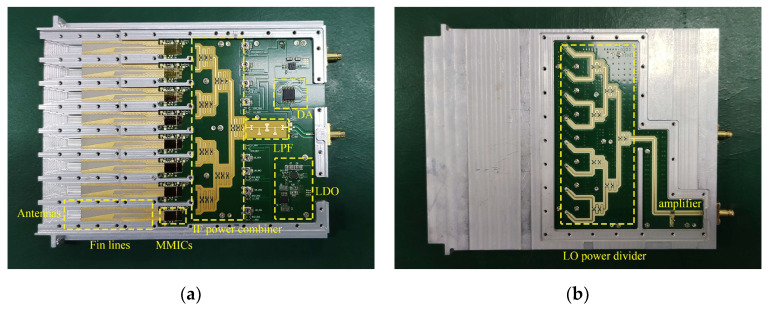
Photograph of the phased array receiver front end: (**a**) front side; (**b**) back side.

**Figure 12 micromachines-14-00859-f012:**
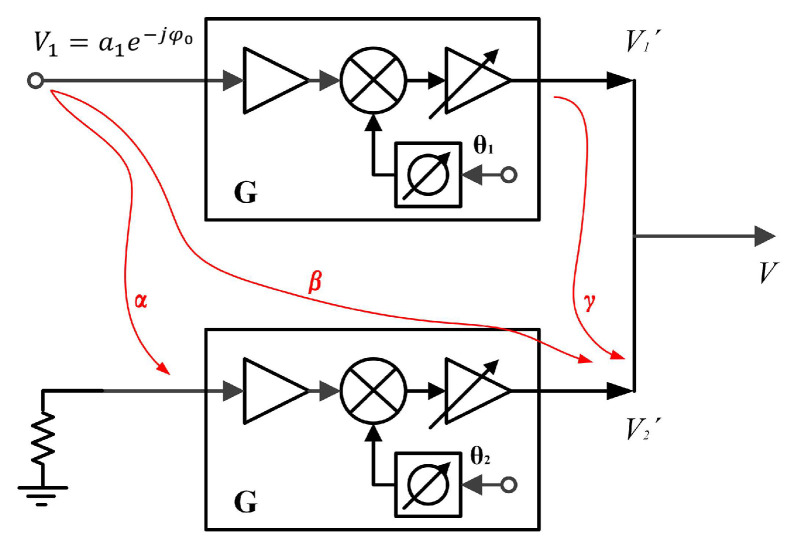
Path analysis of channel mutual coupling.

**Figure 13 micromachines-14-00859-f013:**
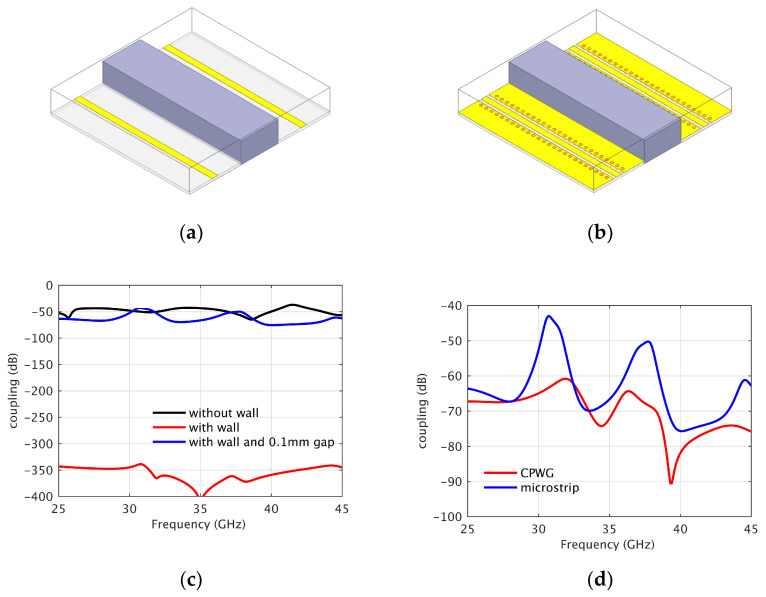
(**a**) Two channels microstrip line model. (**b**) Two-channel CPWG line model. (**c**) Simulation results of microstrip line model in different states. (**d**) Comparison of simulation results between CPWG and microstrip line model.

**Figure 14 micromachines-14-00859-f014:**
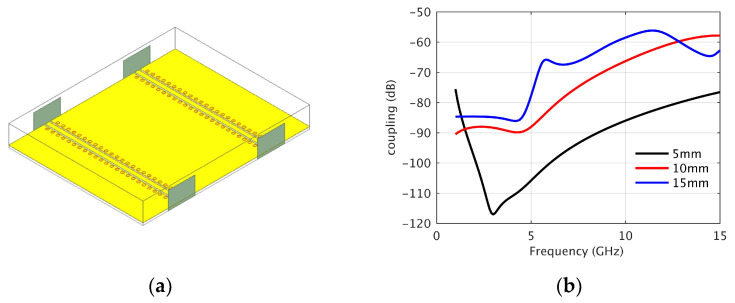
(**a**) Two-channel IF transmission line model. (**b**) Simulation results in different lengths.

**Figure 15 micromachines-14-00859-f015:**
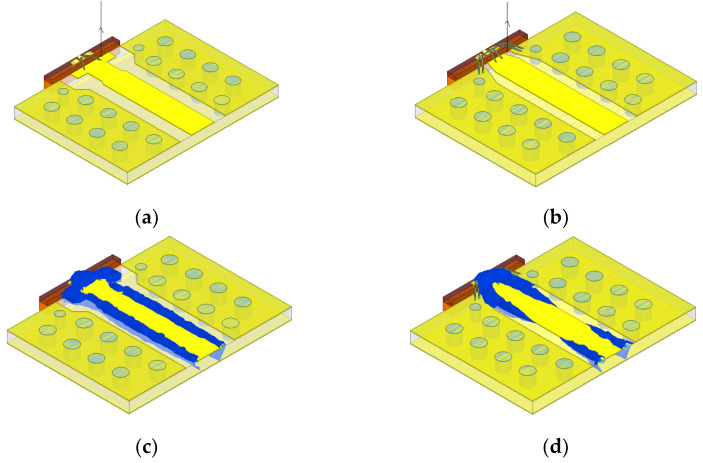
(**a**) Matching compensation bonding wire structure model. (**b**) GSG bonding wire structure model. (**c**) Electric field of matching compensation bonding wire structure. (**d**) Electric field of GSG bonding wire structure.

**Figure 16 micromachines-14-00859-f016:**
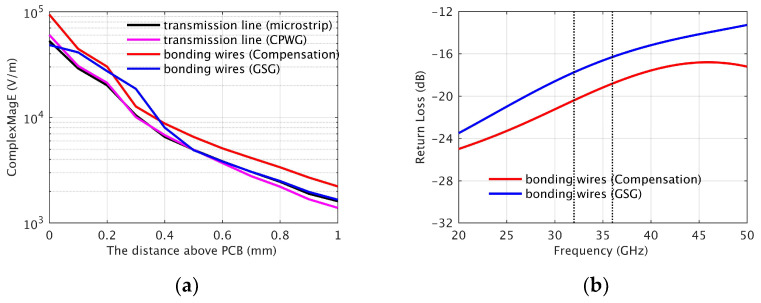
(**a**) Comparison of spatial radiation of different transmission lines and bonding lines. (**b**) Return loss of bonding lines.

**Figure 17 micromachines-14-00859-f017:**
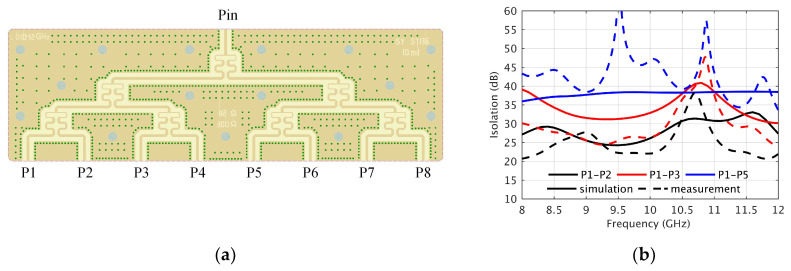
(**a**) LO power divider layout. (**b**) Simulation and measurement results of LO power divider isolation.

**Figure 18 micromachines-14-00859-f018:**
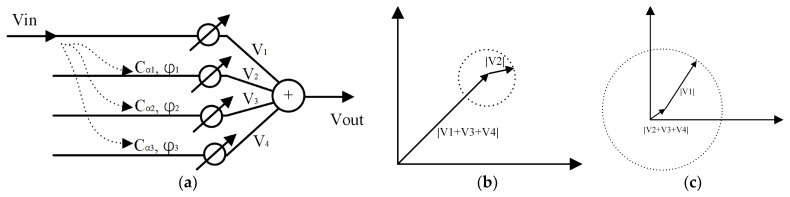
(**a**) Measurement model of channel mutual coupling. (**b**) Changing model of Vout when changing the phase of channel 2. (**c**) Changing model of Vout when changing the phase of channel 1.

**Figure 19 micromachines-14-00859-f019:**
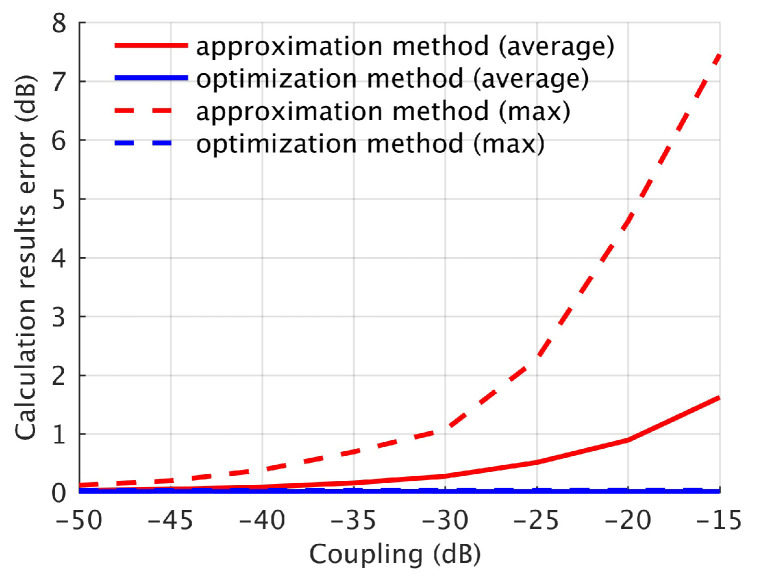
Comparison of coupling errors calculated by approximation method and optimization method.

**Figure 20 micromachines-14-00859-f020:**
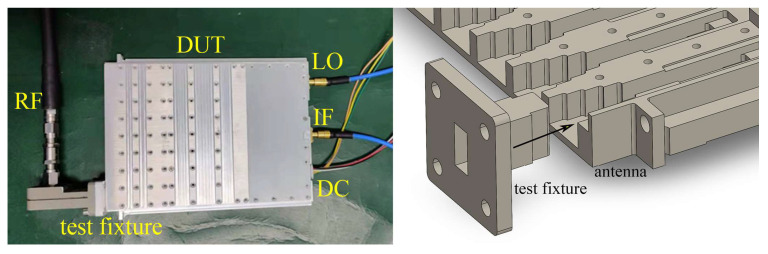
Diagrams of connection mode and test fixture.

**Figure 21 micromachines-14-00859-f021:**
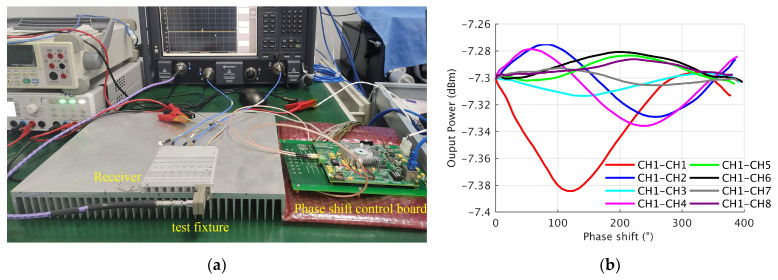
(**a**) The Channel mutual coupling measurement platform. (**b**) Changing curves of the output signal amplitude with phase shift.

**Figure 22 micromachines-14-00859-f022:**
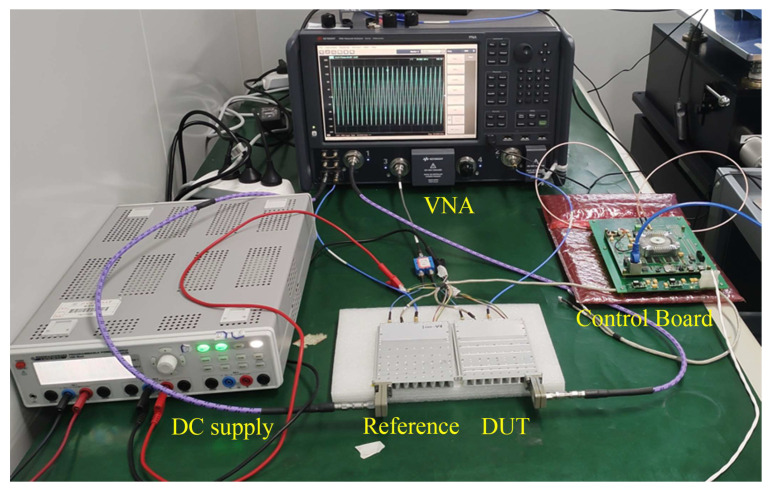
The channel phase error measuring platform.

**Figure 23 micromachines-14-00859-f023:**
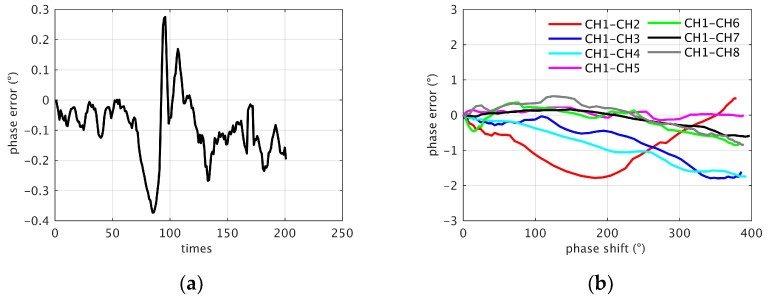
(**a**) Phase measurement error. (**b**) Phase error of channel 1 caused by phase shifter of other channels.

**Figure 24 micromachines-14-00859-f024:**
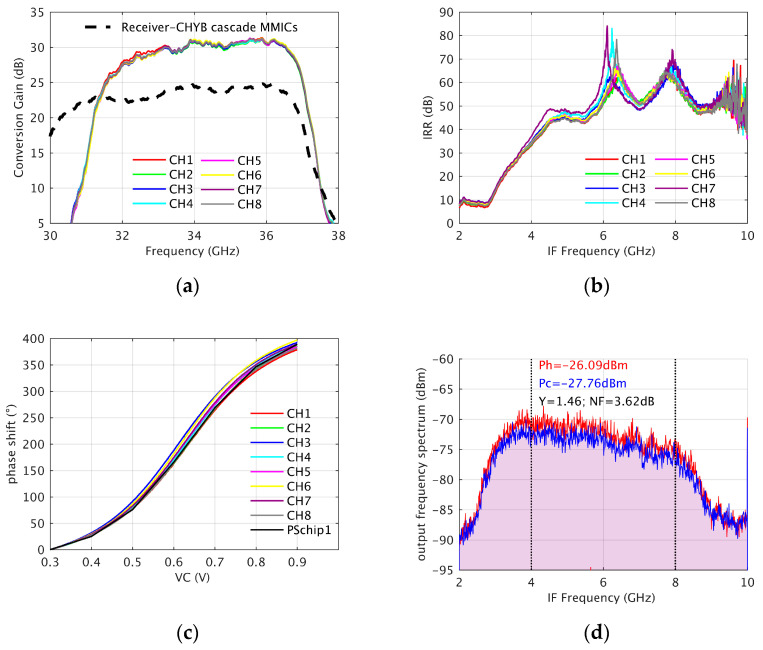
(**a**) Frequency conversion gain measurement results. (**b**) Image rejection ratio measurement results. (**c**) Phase shift range measurement results. (**d**) Noise figure measurement results.

**Figure 25 micromachines-14-00859-f025:**
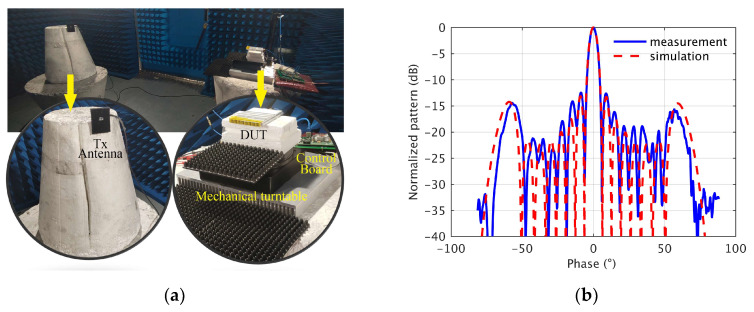
(**a**) The pattern measurement platform. (**b**) The normalized pattern simulation and measurement results.

**Figure 26 micromachines-14-00859-f026:**
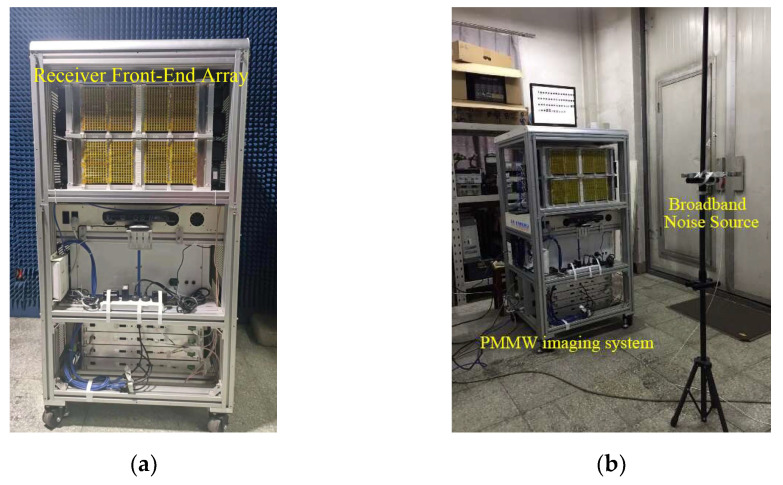
(**a**) The assembled 1024-channel PMMW imaging system. (**b**) The spatial resolution measurement environment.

**Figure 27 micromachines-14-00859-f027:**
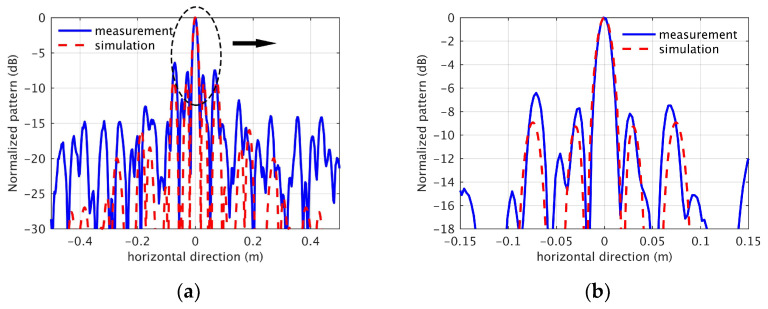
(**a**) The measurement and simulation results of normalized pattern at 1 m in front of the array. (**b**) The main lobe in the pattern.

**Figure 28 micromachines-14-00859-f028:**
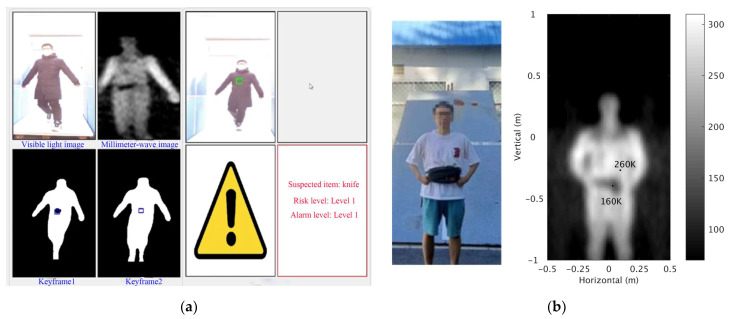
(**a**) Screenshots of the system software. (**b**) Imaging results of the inspected person with metal gun model.

**Table 1 micromachines-14-00859-t001:** Results related to channel mutual coupling.

Channel Number	1	2	3	4	5	6	7	8
**1**	-	−48.9	−67.9	−62.0	−77.2	−62.3	−69.0	−71.7
**2**	−64.3	-	−47.4	−70.6	−61.3	−76.1	−67.9	−68.3
**3**	−62.6	−59.9	-	−48.7	−68.3	−69.9	−66.8	−74.4
**4**	−67.2	−75.9	−65.2	-	−58.3	−66.4	−70.3	−65.4
**5**	−69.6	−61.0	−64.5	−74.5	-	−52.1	−68.1	−64.5
**6**	−65.6	−73.4	−74.2	−68.8	−59.7	-	−55.0	−67.0
**7**	−65.1	−66.4	−63.4	−67.7	−58.0	−56.4	-	−52.8
**8**	−68.2	−70.0	−68.9	−69.4	−61.3	−68.6	−49.9	-

**Table 2 micromachines-14-00859-t002:** Resolution measurement results of the system at different detection distances.

Detection Distance	Imaging Resolution	
(m)	(cm)	(°)
1	1.9	1.09
2	3.72	1.07
3	5.33	1.02
4	7.16	1.03
5	8.82	1.01

**Table 3 micromachines-14-00859-t003:** A comparison of our model with previously published phased array receiver front end.

Ref.	Integration Level	Technology	RF (GHz)	Channel Number	Channel /CG (dB)	NF (dB)	IRR (dB)	Isolation (dB)	Channel /P_DC_ (mW)	Phase Control (°)
[7] 2008	Single-chip	0.13 µm CMOS	22~34	4	9~12	7.5~8	-	27	30	360 (continuous)
[24] 2009	Single-chip	0.18 µm SiGe	40~45	16	12.5	-	-	30	225	360 (4 bit)
[25] 2020	Single-chip	65 nm CMOS	27~31	8	0~3	4	-	32	5	360 (6 bit)
[26] 2022	SIP	0.25 µm GaAs	14.5~16.5	4	22.5	3.4	-	25	-	360 (6 bit)
[15] 2018	MCM	0.15 µm GaAs	60	1	0	-	30	-	800	360 (5 bit)
[11] 2020	MCM	65 nm CMOS	37~40	16	37	5	28	-	6040	360(4 bit)
[27] 2022	MCM	0.18 µm SiGe	15~57	8	25	4.7~6.2	-	-	242	360 (5 bit)
This work	MCM	0.15 µm GaAs	32~36	8	28~31	3.6	35	47	550	360 (continuous)

## Data Availability

Not applicable.

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
