# Peer review of "Ka Band Low Channel Mutual Coupling Integrated Packaged Phased Array Receiver Front-End for Passive Millimeter-Wave Imaging"

_micromachines, 2023, doi:10.3390/mi14040859_

Round 1
Reviewer 1 Report
The author presented an interesting strategy for PMMW and the result are fruitful however there are also some improvement spaces.
Here are some questions and advice.
Q1. For figure one, the real size should be mapped to the picture.
Q2. The author should compare the quantities of these two couplings. (focus on the main contrast)
Q3. Please explain the coupling C matrix in more detail.
Q4. What is the coupling level in figure 3(b)? What is the frequency point you observed?
Q5. The expression in Figure 16 is incorrect, please modify it.
Reviewer 2 Report
In this manuscript the authors describe the fabrication of integrated receiver front end for millimeter-wave imaging. They study the mutual coupling between channels and apply a technique to suppress the coupling. I believe the manuscript worth publishing in Micromachines. For the benefit of the reader, however, several points need clarifying and certain statements require further justification.
1. Page 4, Line 123: It is described that “The near-field pattern ----“. Since the size of each phased array (antenna aperture) will be ~70 mm, the distance of 2 m from the antenna could correspond to far-field region. Please discuss this point.
2. Page 13, Lines 341-342: This sentence is difficult to understand and has grammatical error. Please correct this.
3. Page 13, Equations (11) - (13): These equations are confusing. What are the values of . Please explain in the sentence.
4. Page 17, Figure 25 (a): Please explain more about this figure in the text.
5. Page 18, Figure 27 (a): There exists large discrepancies between experiment and simulation results at the outer region of the horizontal direction, although they agree well in the inner region. Please discuss this point.
6. Page 19, Figure 28. Imaging result of figure 28 (a) seems t show better resolution compared to that of (b). The spatial resolution seems to be worse than 1.85 cm.
